# Targeting Protein Translation in Melanoma by Inhibiting EEF-2 Kinase Regulates Cholesterol Metabolism though SREBP2 to Inhibit Tumour Development

**DOI:** 10.3390/ijms23073481

**Published:** 2022-03-23

**Authors:** Saketh S. Dinavahi, Yu-Chi Chen, Raghavendra Gowda, Pavan Kumar Dhanyamraju, Kishore Punnath, Dhimant Desai, Arthur Berg, Scot R. Kimball, Shantu Amin, Jin-Ming Yang, Gavin P. Robertson

**Affiliations:** 1Department of Pharmacology, The Pennsylvania State University College of Medicine, Hershey, PA 17033, USA; saketh.dvs@gmail.com (S.S.D.); yzc116@psu.edu (Y.-C.C.); cdrgowda@gmail.com (R.G.); pdhanyamraju@pennstatehealth.psu.edu (P.K.D.); kishoresbioworld@gmail.com (K.P.); ddesai@pennstatehealth.psu.edu (D.D.); samin@pennstatehealth.psu.edu (S.A.); 2The Melanoma Center, The Pennsylvania State University College of Medicine, Hershey, PA 17033, USA; 3The Melanoma Therapeutics Program, The Pennsylvania State University College of Medicine, Hershey, PA 17033, USA; 4Department of Public Health Sciences, The Pennsylvania State University College of Medicine, Hershey, PA 17033, USA; aberg1@pennstatehealth.psu.edu; 5Department of Cellular and Molecular Physiology, The Pennsylvania State University College of Medicine, Hershey, PA 17033, USA; srk2@psu.edu; 6Department of Cancer Biology and Toxicology, Markey Cancer Center, College of Medicine, University of Kentucky, Lexington, KY 40536, USA; jyang@uky.edu; 7Department of Pathology, The Pennsylvania State University College of Medicine, Hershey, PA 17033, USA; 8Department of Dermatology, The Pennsylvania State University College of Medicine, Hershey, PA 17033, USA; 9Department of Surgery, The Pennsylvania State University College of Medicine, Hershey, PA 17033, USA; 10Foreman Foundation for Melanoma Research, The Pennsylvania State University College of Medicine, Hershey, PA 17033, USA

**Keywords:** melanoma, EEF2K, cholesterol, SREBP2, HMGCR, LDLR, nanotechnology, anti-cancer drug, drug development

## Abstract

Decreasing the levels of certain proteins has been shown to be important for controlling cancer but it is currently unknown whether proteins could potentially be targeted by the inhibiting of protein synthesis. Under this circumstance, targeting protein translation could preferentially affect certain pathways, which could then be of therapeutic advantage when treating cancer. In this report, eukaryotic elongation factor-2 kinase (EEF2K), which is involved in protein translation, was shown to regulate cholesterol metabolism. Targeting EEF2K inhibited key parts of the cholesterol pathway in cancer cells, which could be rescued by the addition of exogenous cholesterol, suggesting that it is a potentially important pathway modulated by targeting this process. Specifically, targeting EEF2K significantly suppressed tumour cell growth by blocking mRNA translation of the cholesterol biosynthesis transcription factor, sterol regulatory element-binding protein (SREBP) 2, and the proteins it regulates. The process could be rescued by the addition of LDL cholesterol taken into the cells via non-receptor-mediated-uptake, which negated the need for SREBP2 protein. Thus, the levels of SREBP2 needed for cholesterol metabolism in cancer cells are therapeutically vulnerable by targeting protein translation. This is the first report to suggest that targeting EEF2K can be used to modulate cholesterol metabolism to treat cancer.

## 1. Introduction

Melanoma is the fifth most common cancer and while localized disease can be effectively treated by surgical resection, metastatic disease has a five-year survival rate of only 23% [1]. Treatment strategies have improved with the use of BRAF/MEK inhibitors and immune checkpoint targeting approaches [2], However, there remains a need to identify additional novel treatment approaches to increase the overall survival of patients, and one strategy might involve targeting cholesterol metabolism [3].

Cholesterol synthesis and uptake into cells are the two major processes controlling cholesterol metabolism in normal and cancer cells [3]. For cholesterol biosynthesis, the major limiting enzymes for cholesterol production are 3-hydroxy-3-methylglutaryl-co-enzyme A reductase (HMGCR) and squalene epoxidase (SQLE) [4]. However, the use of statins, such as simvastatin or lovastatin, to control cholesterol synthesis appears to have a negligible effect on cancer prevention or development in animal models and in patients [3]. To unravel this discrepant observation, the other process key to cholesterol metabolism has been studied. A more prominent process to regulate cancer development appears to involve targeting the uptake and transport of cholesterol into cancer cells from the extracellular environment [5,6,7,8]. Targeting these processes appears quite effective for controlling the growth as well as progression of cancer in preclinical models and clinical testing of this approach is now occurring [6,8,9]. Thus, cholesterol uptake and transport into cancer cells appears to be more important than cholesterol synthesis, making targeting of the former more critical than the later process [5,6,7,8].

For cholesterol uptake, low density lipoproteins (LDL) are the major sources transported into the cancer cell [3]. The LDLs uptake can occur by first binding to LDL-receptors (LDLR) followed by receptor mediated endocytosis to release LDL into the endosomes [9]. Alternative approaches involving non-receptor-mediated-uptake, such as occurring through micro and macro-pinocytosis have also been reported and are being investigated in cancer [10,11]. Internalized LDLs are degraded in lysosomes, while cholesterol is transported to the cytoplasm through Niemann–Pick type C1 (NPC1) protein [12]. Free cholesterol enters the cellular membrane pool via the endoplasmic reticulum (ER), or is converted into steroid hormones in mitochondria, or is stored as cholesterol esters in cytoplasmic lipid droplets [13]. It has been reported that targeting NPC1 using Leelamine reduced cholesterol metabolism and melanoma tumour development by disrupting the transport or release of external cholesterol from lysosomes into the cellular pool [6].

One underexplored area of research to modulate cholesterol metabolism involves targeting the protein translational machinery involved in the production of proteins for this pathway. Then to show through rescue experiments that cholesterol metabolism in cancer cells is therapeutically vulnerable by targeting protein translation. To unravel this process, several enzymes involved in protein translation were evaluated as potential modulators of cholesterol uptake. Eukaryotic elongation factor-2 kinase (EEF2K) was identified as a potential key regulator of both cholesterol uptake and synthesis. The enzyme EEF2K, also known as calmodulin-dependent protein kinase III, is a unique calcium/calmodulin-dependent enzyme that regulates protein translation [14,15,16]. It phosphorylates EEF2 on Thr-56 to inactivate this elongation factor, which blocks translational elongation to inhibit protein synthesis [15]. As protein translation is an energy- and resource-consuming cellular process [17], it is not surprising that EEF2K is overexpressed in cancers as a mechanism to increase protein synthesis [17]. Downregulation of EEF2K has been found to selectively decrease the levels of “pro-oncogenic” proteins, causing growth inhibition and the induction of apoptosis [18]. The NH125 compound is a widely-reported EEF2K inhibitor, which decreases cancer cell viability [19] by increasing EEF2 phosphorylation [20,21] to reduce mRNA translation into protein, leading to inhibition of tumour growth [21].

While it is known that EE2FK regulates protein synthesis, this report examines the novel role it plays in regulating cholesterol metabolism in melanoma by modulating the translation of SREBP2 mRNA into protein. The expression of EEF2K was found to be important for melanoma cell proliferation, by regulating the production of proteins important in the uptake of cholesterol, compared to other translated proteins. Specifically, this process was mediated through translation of SREBP2 mRNA, the key cholesterol biosynthesis transcription factor. When EEF2K was targeted, it reduced the levels SREBP2 to inhibit cholesterol metabolism. Importantly, rescue of this process by the addition of LDL cholesterol to the cells, suggests the importance of protein translation in the functioning of the cholesterol pathway in cancer cells, compared to other proteins translated in melanoma.

## 2. Results

### 2.1. Knockdown of EEF2K Expression Inhibits Melanoma Cell Proliferation

To examine the role of EEF2K in melanoma, the expression levels and corresponding survival of melanoma patients was examined using the TCGA database (data assessed through the Xena browser). While not significant (*p* = 0.2194), a trend is apparent that higher expression of EEF2K tend to be associated with a poorer prognosis (Figure 1A). Patients designated as high EEF2K (*n* = 105) are the top 25% quartile, while those listed as low (*n* = 115), were in the bottom 25% quartile of the TCGA database. Next, levels of EEF2K were knocked down in melanoma cell lines using four different siRNAs (Figure 1B, C) and effects on cellular proliferation examined by assessing metabolism using the MTS assay. All four siRNAs targeting EEF2K reduced proliferation of 1205 Lu melanoma cells (Figure 1D, *p* < 0.0001). The siRNA #1 targeting EEF2K was further evaluated at different concentrations for an effect on cellular proliferation. A dose-dependent decrease in the proliferation of both 1205 Lu (Figure 1E, *p* < 0.0001) and UACC 903 (Figure 1F, *p* < 0.0001) cells was observed following knockdown of EEF2K. Additionally, EEF2K protein amounts were reduced to undetectable levels in melanoma cell lines transfected with 200 pmole of EEF2K-targeted siRNA #1 (Figure 1B). Knockdown of BRAF protein served as a positive control, and the results were compared to knockdown with a negative scrambled siRNA control. The reduction in cellular proliferation when targeting EEF2K was similar to that observed when the BRAF (positive control) was knocked down (Figure 1D–F), suggesting that EEF2K might be an important target in melanoma. The use of BRAF siRNA serves as a control for assessing effects on cellular proliferation have been published previously [22].

### 2.2. Knockdown of EEF2K Reduced Cholesterol Levels by Modulating Levels of SREBP2 Protein

Since cholesterol metabolism is considered to be an important pathway involved in melanoma development [3], the effect of inhibiting EEF2K on cholesterol levels in cultured cells was examined. The EEF2K protein was knocked down using siRNA and levels of cholesterol were measured in 1205 Lu cells (Figure 2A, *p* < 0.0001). All four siRNAs evaluated reduced the cholesterol levels in 1205 Lu cells by up to 50%. The siRNA #1 was further evaluated in UACC 903 cells (Figure 2B, *p* < 0.0001), showing a more that 50% reduction in cholesterol levels. Western blot analysis showed that knockdown of EEF2K using siRNA #1 decreased the protein levels of SREBP2 (transcriptional regulator of LDL-R and HMGCR pathways), and the proteins it modulates, LDL-R (crucial for cholesterol uptake) and HMGCR (rate limiting enzyme for cholesterol synthesis), in UACC 903 cells (Figure 2C).

### 2.3. The Cholesterol Metabolism Was Regulated by EEF2K through the Reduction of LDL-R and HMGCR

To demonstrate that cholesterol metabolism is an important process regulated by EEF2K (as the translation of many proteins would likely be similarly affected), we examined whether the effect of targeting EEF2K on cholesterol levels could be rescued by addition of LDL bound cholesterol or mevalonic acid (the product of HMGCR enzyme reaction important for cholesterol synthesis). Mechanistically, blocking receptor-mediated uptake of LDL can be overcome by adding excess LDL, which is then taken into the cells by non-receptor-mediated-uptake, through micro and macro-pinocytosis [10,11]. Complete rescue of cellular cholesterol levels was observed in UACC 903 (Figure 2D, *p* < 0.0001) and 1205 Lu (Figure 2E, *p* < 0.0001) cells by adding LDL into the media. In contrast, cholesterol levels were not rescued as effectively in cells treated with mevalonic acid (Figure 2D and Figure 2E, *p* = 0.0483 and *p* = 0.0461 respectively), suggesting that cholesterol uptake into the cells might play a more important role than cholesterol synthesis. Rescue of proliferation of UACC 903 (Figure 2F, *p* < 0.0001) and 1205 Lu (Figure 2G, *p* < 0.0001) cells following siRNA-mediated targeting of EEF2K were observed by the addition of LDL and mevalonic acid. This observation was further investigated by knockdown of the cholesterol pathway genes SREBP2, HMGCR and LDLR and examination of effects of cell proliferation. Only knockdown of the transcription factor, SREBP2 which regulated both, inhibited cell proliferation. This suggested that SREBP2 might play an important role in cell survival (Appendix A, *p* < 0.0001).

The compound NH125 (Figure 3A), which not only is an EEF2K inhibitor but can promote EEF2 phosphorylation to regulate protein translation [20,21], demonstrated a similar pattern of inhibiting both cholesterol metabolism and proliferation of melanoma cells (Figure 3). While other EEF2K inhibitors have been reported, NH125 is currently the most effective agent [15,23,24,25,26] and was therefore selected for this study. The A484954 EEF2K inhibitor [27] did not reduce the levels of pEEF2K or inhibit proliferation of melanoma cells even at 100 µM concentrations and was therefore not further evaluated in this study. As reported for other cancer types [15,23,24,25,26], the IC_50_ of NH125 was similar for melanoma cells, ranging from 1.5 µM to 3 µM (Figure 3B). The agent NH125 potently inhibited cellular proliferation of fibroblasts (MRC5 and FF2441) and melanoma cell lines (1205 Lu, UACC 903, A375M, vemurafenib-resistant A375M (A375MR), C8161 Cl9 and A2058), when using an MTS assay. Targeting EEF2K with NH125 reduced the phosphorylation of EEF2K protein in UACC 903 cells, suggesting that inhibiting its activity is causally associated with suppression of melanoma cell growth (Figure 3C). Furthermore, 5 μM NH125 reduced cholesterol levels in UACC 903 and 1205 Lu cells by approximately 50% after 24 h exposure (Figure 3D, E, *p* < 0.0001). The reduction in cholesterol levels were not significantly different between NH125 and lovastatin (potent HMGCR inhibitor) treated UACC 903 (Figure 3D, *p* < 0.0001) or 1205 Lu (Figure 3E, *p* < 0.0001) cells. Importantly, inhibiting cholesterol synthesis using Lovastatin did not have any effect on melanoma cell survival up to 20–30 µM (Appendix A). The NH125 compound dose-dependently reduced LDL-R, HMGCR, SREBP2 and SREBP1 levels in UACC 903 cells to decrease cholesterol metabolism (Figure 3F).

### 2.4. The EEF2K-Regulated Cholesterol Metabolism Was Mediated by SREBP2 and LDL-R 

To demonstrate that the effect of NH125 on cellular proliferation was mediated through modulation of cholesterol uptake, rescue experiments were undertaken with LDL added to the culture medium to reverse the growth inhibitory effects mediated by NH125 on EEF2K. The 100 µg/mL LDL rescued UACC 903 cells from NH125-induced cell death at 1.25 µM (Figure 4A, *p* < 0.0001), while there was no effect with addition of mevalonic acid (Figure 4B, *p* = 0.9622–0.9756). Similarly, there was no rescue of cholesterol levels in cells treated with mevalonic acid, but the addition of LDL increased the cholesterol levels to close to those observed in the controls (Figure 4C, NH125 vs. NH125 + LDL: *p* < 0.0001, NH125 vs. NH125 + mevalonic acid: *p* = 0.281).

To identify the key proteins in the cholesterol pathways underlying this effect, SREBP2, HMGCR and LDLR were knocked down using siRNA, and effect on cholesterol levels as well as killing these cells by NH125 was examined. The agent NH125 did not reduce the level of cholesterol in UACC 903 cells in which SREBP2 or LDLR had been knocked down, but it did in cells in which HMGCR had been targeted (Figure 4D, *p* = 0.97 and 0.89 respectively). Furthermore, cellular proliferation was only affected in UACC 903 cells in which HMGCR, had been targeted (Figure 4E, NH125 treatment in siScramble vs. in siREBP2: *p* < 0.0001, vs. siLDLR: *p* < 0.0001, vs. siHMGCR: *p* = 0.98). Thus, eliminating SREBP2 and LDLR prevented the effects of NH125 on the cellular cholesterol levels and on proliferation. The effects of NH125 were retained only when HMGR was knocked-out, suggesting the signalling was mediated through SREBP2 and LDLR.

### 2.5. Modulation of the Translation of Proteins Important in the Cholesterol Metabolism Pathway by Targeting EEF2K 

Since EEF2K is a regulator of translational elongation of proteins [14,15], the effect of EEF2K inhibition on the translation of genes essential in the cholesterol pathway was evaluated. To dissect the mechanisms by which EEF2K regulates cholesterol gene translation, a polysomal analysis was performed [28]. Polysomal analysis is a technique to evaluate the association of mRNA with the ribosomes/polysomes and identify active polysomes translating mRNA and ribosomes that are not [28]. UACC 903 cells treated with 5 µM NH125 for 24 h had significantly reduced levels of protein translation, as evidenced by a reduction in the number of ribosomes in the polysomal fractions of the cell lysates (Figure 5A). In contrast, the 80S fraction of the NH125-treated cells was higher while the corresponding polysomal fraction was lower as compared to the DMSO-treated cells. The ribosome and polysome fractions were then collected and subjected to PCR analysis to identify the specific mRNA of cholesterol genes undergoing translation. SREBP2 and LDL-R mRNA were found to be present in the 80S ribosomal fraction in the DMSO control but absent from the polysomal fraction in the presence of NH125 (Figure 5B). Collectively, these data suggest that NH125 modulates cholesterol metabolism by reducing the initiation of translation of the SREBP2 transcription factor and its target (LDL-R) into proteins.

### 2.6. Development of a Bioavailable Less Toxic Nanoliposomal Formulation of NH125

Targeting the translational elongation by inhibiting EEF2K/EEF2 proteins in animals is limited by availability of a non-toxic drug [14,15,16]. The EEF2K inhibitor NH125 is currently the most effective EEF2K inhibitor [15,23,24,25,26] but it is toxic with a maximum tolerated dose (MTD) in Swiss Webster mice of 0.5 mg/kg, higher doses are lethal (Appendix A). At 1 mg/kg, NH125 significantly reduced animal body weights causing hunched backs and a distorted abdomen. Therefore, the maximum tolerated dose of NH125 without any toxic effects was 0.5 mg/kg. 

To increase bioavailability of NH125 without toxicity, a nanoliposomal formulation, termed NanoNH125 was developed. The maximum tolerated dose of NanoNH125 was 8-fold higher than NH125, indicating the potential effectiveness of encapsulating and controlling the release of the drug, in order to reduce the toxicity (Appendix A). The encapsulation efficiency was found to be 68% (Appendix A) and average size and charge of the nanoliposomes was ~70 nm and −2.5 eV, respectively. The size (Appendix A, *p* = 0.99), charge (Appendix A, *p* = 0.99) and activity (Appendix A, *p* = 0.97) of the nanoliposomes did not change when stored at 4 °C for up to 6 months, indicating the stability of the formulation.

### 2.7. Targeting EEF2K with siRNA and NanoNH125 Inhibited Tumour Growth and Cholesterol Levels with Manageable Toxicity

To determine whether targeting EEF2K would affect melanoma tumour development similar to what we have observed in breast cancer [14] when EEF2K was knocked down, an established rapid approach was used [29,30,31], which involved nucleofecting UACC 903 melanoma cells with 200 pmole of EEF2K, BRAF or scrambled siRNAs and 24 h later, injecting these cells subcutaneously into the flanks of nude mice. The result of knocking down BRAF expression and using it as an in vivo control have been published [22]. Tumour growth was measured on alternate days, showing that knockdown of EEF2K reduced the tumour growth by up to 65%, in a manner similar to cells in which BRAF had been targeted and compared to the siScramble control (Figure 6A, *p* < 0.0001). As required for this assay, knockdown of EEF2K protein persisted for up to the required eight days following siRNA nucleofection with EEF2K [22] in UACC 903 cells (Figure 6B). Knockdown of BRAF expression for its use as a positive control has been reported previously [22].

To compare the anti-tumour efficacy of NanoNH125 versus non-encapsulated NH125, mice bearing vascularized melanoma xenografts (7 days after cell implantation) were initially treated with 0.5 mg/kg NH125 in DMSO. Figure 6C shows that the free NH125 reduced UACC 903 xenografts size by an average of 34% after Day 28, compared to the DMSO control (*p* < 0.0001). Since the higher doses of free NH125 were toxic in animals (Appendix A), they could not be evaluated. In contrast, NanoNH125 was evaluated from 0.5 to 8 mg/kg body weight and only at 4 mg/kg was toxicity observed (Appendix A). Therefore, NanoNH125 tumour inhibitory efficacy was assessed at 1 and 2 mg/kg (Figure 6D, *p* < 0.0001). The treatment caused a dose-dependent reduction in tumour size with no significant effect on animal weight or behaviour (Figure 6D,E insets, and Appendix A). The NanoNH125 at 2 mg/kg reduced the tumour volumes by approximately 65%, compared to empty liposome treated controls. Similarly, tumour inhibitory efficacy of 72% was observed in the 1205 Lu xenograft model (Figure 6E, *p* < 0.0001).

To assess the tumour inhibitory efficacy of free NH125 versus NanoNH125 in an unbiased manner, the DMSO and EL groups were combined and used as a single control to which the treatments were compared and subjected to statistical analysis (Appendix A). The 0.5 mg/kg NH125 treatment led to a tumour reduction of 29.4%; the 1 mg/kg NanoNH125 treatment reduced tumours by 49.8%; and the 2 mg/kg NanoNH125 treatment inhibited tumour growth by 70.7%. Evaluation of changes in tumour kinetics showed that the 0.5 mg/kg NH125 treatment only delayed tumours reaching an average size of 750 mm^3^ by 2 days (Appendix A). In contrast, when mice were treated with 2 mg/kg NanoNH125, 50% of UACC 903 and 67% of 1205 Lu tumours did not reach an average of 750 mm^3^ by the end of experimentation (Appendix A). Thus, there was a statistically significant benefit for using NanoNH125 over free NH125.

NanoNH125 also reduced the cholesterol levels in the tumours when compared to empty liposome controls (Figure 6F, *p* < 0.0001). Thus, these results demonstrated the successful development of a new formulation of the EEF2K inhibitor, NH125, for modulating cholesterol metabolism and inhibiting tumour growth, which has less toxicity than NH125 in DMSO.

## 3. Discussion

Cholesterol is an essential component of cellular membranes, playing a key role in the integrity and function of normal and cancer cells [3]. Highly proliferative cancer cells increase the uptake of exogenous (or dietary) lipids (including cholesterol) and lipoproteins instead of increasing synthesis, which makes targeting this process potentially important for cancer management [3]. Excess lipids and cholesterol in cancer cells are stored in lipid droplets (LDs) [3]. High levels of LDs and stored-cholesteryl ester content in tumours are considered as hallmarks of cancer aggressiveness [3]. Moreover, LD-rich cancer cells are more resistant to chemotherapy [3]. Since cholesterol is needed for normal cellular function, it is synthesized de novo from acetyl-CoA in the mevalonate pathway [3]. Therefore, it is possible that therapeutic strategies designed to target cholesterol uptake would be less harmful to normal cells producing it, thereby be a useful therapeutic strategies to kill cancer cells.

Targeting the lipid and cholesterol uptake processes have been shown to be effective at reducing tumour development [6,8,9]. Inhibiting cholesterol transport protein NPC1 using Leelamine led to a significant reduction in melanoma growth and metastasis [5,6]. Similarly, compounds that change the pH of lysosomes to disrupt cholesterol metabolism from LDL, namely lysosomotropic compounds, have been shown to inhibit tumour development with manageable toxicities [8]. However, there exists a critical need to identify new targets and approaches to inhibit cholesterol uptake to better manage cancer progression. This is the first report suggesting that EEF2K modulates the translation of cholesterol metabolism-regulating proteins from mRNA. Furthermore, targeting this protein could be used to modulate tumour development by decreasing cholesterol metabolism [3].

The compound EEF2K is a protein synthesis elongation factor overexpressed in various types of cancer, which plays important roles in activating autophagy, maintaining cellular energy and supporting tumour cell survival under various stressful conditions [14,15]. Here, the importance of EEF2K in the proliferation of melanoma cells is demonstrated. Genetic knockdown of EEF2K significantly melanoma cell growth in culture and as tumours in animals in a manner similar to that observed following the knockdown of ^V600E^BRAF protein, which suggested the potentially important role of translational elongation in the survival of melanoma cells [15].

Targeting EEF2K, or its downstream substrate EEF2, using siRNA or NH125 reduced the levels of LDL-R, HMGCR and SREBP2, which led to a significant reduction of cholesterol levels in cells [3,4,13]. It is also possible that other factors regulating SREBP2 maturation, such as increased levels of Sterol Regulatory Element Binding Protein Cleavage Activating Protein (SCAP) [32,33] resulting from changes in protein translation could be involved in this process. The protein SCAP induces translocation of SREBP from the endoplasmic reticulum to the Golgi apparatus, allowing it to regulate cellular cholesterol levels [34]. The possible involvement of SCAP in this process was not investigated. Future studies could also investigate the effect of modulating EEF2K and EEF2 with NanoNH125 on the immune system in immunocompetent models as cholesterol metabolism can play an important role in immune cells [35].

Cellular proliferation and cholesterol levels were rescued by external addition of cholesterol (LDL) validating the significant role of cholesterol uptake in this process. Because receptor-mediated uptake is blocked, cholesterol was taken into the cells by non-receptor-mediated-pinocytosis to overcome the effects [10,11]. This is likely what is occurring in mice in which EEF2K is knocked out to enable survival since they have an almost normal phenotype. They likely obtain LDL cholesterol from their diet to compensate [36,37]. In contrast, there was only a partial rescue with addition of mevalonic acid, the precursor for cholesterol biosynthesis [4]. Since the rescue was not as prominent as seen with addition of LDL, a possible interpretation could be that the cholesterol uptake process is more important for the survival of melanoma cells than cholesterol biosynthesis. This possibility is supported by previous reports demonstrating the importance of cholesterol uptake compared to cellular synthesis for the modulation of melanoma development [5,6,7,8].

Knockdown of EEF2K in melanoma cells reduced the levels of cholesterol and the protein levels of LDL-R (crucial for cholesterol uptake), HMGCR (rate limiting enzyme for cholesterol synthesis), and SREBP2 (transcriptional regulator of both pathways). It is likely that the levels of many proteins were similarly affected but the importance of the cholesterol pathways in this process was demonstrated by reversing the effects through the addition of LDL cholesterol. The LDL addition led to complete rescue when EEF2K was inhibited using siRNA or with the EEF2K inhibitor, NH125. Furthermore, knockdown of SREBP2 or LDL-R level in the cells reduced the effect of NH125 on the cholesterol levels in tumour cells, suggesting that these proteins are important mediators of the effects modulated by NH125. Finally, decreased polysome association of the SREBP2 and LDL-R mRNAs encoding genes important in cholesterol metabolism with a concomitant increased association with the 80S ribosomal fraction was observed, suggesting that the inhibition of the translation of proteins important in cholesterol metabolism occurred. However, the polysome analysis after inhibition with NH125 is suggestive of inhibition of translational initiation, which could be considered inconsistent with EEF2K being an elongation factor [38]. This is expected since NH125 causes increased eIF2α phosphorylation [39], and slowing protein elongation using cycloheximide has been reported to increase eIF2α phosphorylation leading to the subsequent suppression of translational initiation [40]. Therefore, it is reasonable to expect that NH125 would block the initiation and elongation involved in protein translation.

Another important aspect in intracellular cholesterol homeostasis is cholesterol movement and storage in cells through the SREBP2-regulated generation of liver-X-receptors (LXR) ligands [41]. These LXRs act as ligand-activated transcription factors [42], binding to the promoter region of ATP-binding cassette transporters (ABC transporters), leading to the expression of ABCA1 and ABCG1 [42]. ABCA1 and ABCG1 can control the levels of cholesterol in-and-out of cells [43]. Little is known about the effect of EEF2K inhibition on the LXR ligands but SREBP2 can increase ABCA1 expression by maintaining the supply of endogenous oxysterol ligands for LXR through the mevalonate pathway [41]. Therefore, when SREBP2 is inhibited, cholesterol uptake and efflux could be reduced through decreased LDL-R and ABCA1 to preserve the intracellular cholesterol pool. This possibility is further supported through a study that showed deletion of ABCA1 in myeloid cells prevented melanoma tumour growth in synergetic C57/B6 mice [44]. While not explored in this project, it is possible that targeting EEF2K could also be reducing ABCA1 levels, in addition to LDL-R, leading to the observed effects.

The mechanisms used by EEF2K/EEF2 to affect protein translation and cell survival are diagrammed in Figure 7. Active EEF2K phosphorylates EEF2 and inhibits protein translation, which leads to energy and resource conservation and favoured pro-survival protein synthesis. These steps subsequently promote cell survival (Figure 7 left panel). Targeting EEF2K with siRNA and with the EEF2k inhibitory agent, NH125 regulated protein translation mediated by EEF2 and reduced protein synthesis. Our results indicated that EEF2K regulated the expression levels of SREBP2 and its downstream targets, LDLR and HMGCR which play important role in cholesterol metabolism in cancer cells and contributes to cell survival. In contrast, genetically and pharmacologically inhibiting EEF2K reduced the levels of SREBP2, LDLR, HMGCR and cellular cholesterol to inhibit cell proliferation.

The inhibitory effect of NH125 has been examined on multiple cultured cancer cell types, including those of the oesophagus, brain, ovary, cervix, prostate and breast [15,23,24,26]. Furthermore, NH125 is also somewhat effective in animal models at low non-toxic doses below those used in this study, for cancers of the oesophagus [26], brain [45] and breast [46]. These studies support our observations in melanoma by showing that NH125 can inhibit cancer cell proliferation and tumour. Since NH125 is a toxic agent, it has limited potential as an anti-cancer treatment, therefore, it was formulated into a nanoliposome in an effort to mitigate its toxicity. Incorporating drugs into nanoliposomes can reduce the inherent toxicity due to a more controlled release along with increasing the half-life of drugs [47,48]. The loading efficiency of NanoNH125 was calculated to be 68% and average size and charge of the nanoliposomes was approximately 70 nm and charge of −2.5 eV in saline, respectively. Importantly, the size, charge, and activity of the nanoliposomes did not change when stored at 4 °C for up to 6 months indicating the stability of the formulation. The maximum tolerated dose of NanoNH125 was 8-fold higher than free NH125 demonstrating the success of this approach. More importantly, NanoNH125 significantly reduced the tumour volumes by up to 70.7% at 2 mg/kg dose without apparent toxicities, indicating the successful development of a potent, less-toxic formulation of NH125 to suppress melanoma development. Mice in which EEF2K is knocked out, survive and have an almost normal phenotype as they likely obtain LDL cholesterol from their diet to compensate [36,37]. In contrast, the EEF2K inhibitor, NH125, is toxic to normal mice as they have likely not developed this compensatory mechanism and the disruption of cholesterol metabolism has more drastic effects.

Abnormal protein expression is a hallmark of malignancies and generally associated with a poor prognosis [49]. Proteins are generally expressed that are beneficial to tumour growth and that enable the development of drug resistance [50,51]. Therefore, rather than targeting individual proteins, strategies limiting the process of protein translation to inhibit tumours [49] could be a useful adjunct to current cancer therapies [50,51,52,53]. However, approaches to accomplish this objective are still be investigated in the preclinical models and need a better understanding of the mechanistic underpinning that could make them useful [49]. Since protein synthesis is significantly upregulated in cancer cells, targeting this process could more effectively kill cancer rather than normal cells. This study suggests that targeting EEF2K to control mRNA translation can be used as a unique approach to inhibit cancer by controlling cholesterol metabolism.

## 4. Materials and Methods

### 4.1. Cell Lines and Culture Conditions

Normal human lung fibroblasts (MRC5) were purchased from ATCC and human fibroblasts (FF2441) were provided by Dr. Craig Myers, Penn State College of Medicine, Hershey, PA. The human melanoma cell line 1205 Lu were provided by Dr. Meenhard Herlyn, Wistar Institute, Philadelphia, PA. The human melanoma cell line A375M was obtained from ATCC. The human melanoma cell line UACC 903 was provided by Dr. Mark Nelson, University of Arizona, Tucson, AZ. The wildtype BRAF melanoma cell line C8161.Cl9 was provided by Dr. Danny Welch, University of Kansas, Kansas City. All cell lines were tested for mycoplasma and confirmed to be negative. The STR analysis with comparison to the reference STR profiles confirmed the identity of key cell lines, UACC 903 and 1205 Lu. Cell lines were maintained in a 37 °C humidified 5% CO_2_ atmosphere incubator and constantly monitored for phenotypic and tumorigenic characteristics to further validate cell line identity.

### 4.2. siRNA Transfections

Duplex stealth siRNA sequences for scrambled, EEF2K, LDL-R, HMGCR, SREBP2 and BRAF were purchased from Invitrogen. 50–200 pmole Individual siRNAs were introduced into UACC 903, 1205 Lu, or C8161.Cl9 cells via RNAimax (Invitrogen, Waltham, MA, USA) or by Nucleofection, as published previously [29,54].

Transfection and nucleofection efficiency were >90% with 80–90% cell viability. Transfected or nucleofected cells were plated and allowed to recover for 3 days and then used for experiments unless specified otherwise. This approach has been validated to knockdown protein levels for up to 8 days after nucleofection [29,54].

### 4.3. Cell Viability Assay

Cell viability assays were performed on various cell lines following transfection/ nucleofection or pharmacological agent s as described previously [55,56,57]. Briefly, cells were incubated with various concentrations of NH125 (Tocris, Bristol, UK) or lovastatin (Tocris) for 72 h and treated with MTS for 1-h followed by measuring absorbance at 492 nm. 50–100 μg/mL LDL and 50–100 μM mevalonic acid (Tocris) was added as rescue agents. The IC_50_ values or percentage of cells for each experimental group were calculated in three independent experiments using GraphPad Prism version 7.04.

### 4.4. Western Blot Analysis

Cell lysates were prepared as previously described [58,59]. For siRNA experiments, cell lysates were collected 2, 4, 6 and 8 days after the introduction of siRNA into the cells to validate knockdown of protein levels by western blotting, as previously reported [29,54]. For experiments with NH125, the agent was added after 48 h of incubation and protein lysates collected following 24 h of treatment. Blots were probed with antibodies according to each supplier’s recommendations: antibodies to EEF2K, p-EEF2K, alpha-enolase, LDL-R, HMGCR, SREBP2 and secondary antibodies conjugated with horseradish peroxidase from Santa Cruz Biotechnology. Alpha-enolase served as the control for protein loading.

### 4.5. Cellular Cholesterol Measurement

After a 2-day recovery, cells into which siRNA had been introduced were treated with 5μM NH125 or 20 μM lovastatin for 24 h. 100 μg/mL LDL or 100 μM mevalonic acid was added to the media for 24 h, serving as rescue reagents. Cells were harvested and treated with 2 N KOH solution and heated for 2 h at 70 °C for hydrolysis of lipids. The homogenate was neutralized with HCl and the lipids were extracted by addition of chloroform followed by phase separation. Chloroform was then evaporated under a stream of nitrogen and the lipid layer was reconstituted in a solution of 70:30 Methanol: Acetonitrile. The levels of cholesterol in the lipid solution were estimated using HLPC with a C18 column and 70:30 Methanol: Acetonitrile for isocratic flow solution. Absorbance was measured at 254 nm.

### 4.6. LDL and Mevalonic Acid Rescue of Cell Proliferation Following siRNA or NH125 Targeting of EEF2K

Two days after introduction of siRNA into the cells, the media was supplemented with 50–100 μg/mL LDL or 50–100 μM mevalonic acid for 3 days. For experiments with NH125, cells were treated with NH125 at various concentration simultaneously with LDL or mevalonic acid. Cholesterol levels and cell viability were assessed by the HPLC and MTS assay, respectively.

### 4.7. Polysomal Analysis

Following the treatment with 5 μM NH125 for 24 h, cycloheximide (CHX) (100 μg/mL) was added to cells for 10 min at 37 °C. Cells were washed twice with cold PBS containing CHX (100 μg/mL), scraped off and pelleted at 4000 rpm (1400× *g*) for 10 min at 4 °C. The cell pellets were suspended in 500 μL of lysis buffer (10 mM HEPES-KOH at pH 7.4, 2.5 mM MgCl_2_, 100 mM KCl, 0.25% NP-40, 100 μg/mL CHX, 1 mM DTT, 200 unit/mL RNase inhibitor [RNaseOUT, Invitrogen] and EDTA-free protease inhibitor. Lysates were cleared at 14,000 rpm (18,000× *g*) for 15 min at 4 °C and supernatants (cytosolic cell extracts) were collected and the absorbance at 260 nm was measured. Lysates were layered over 10–50% or 15–50% 4 °C sucrose gradients in buffer (10 mM HEPES-KOH at pH 7.4, 2.5 mM MgCl2, 100 mM KCl). Gradients were centrifuged at 17,000 rpm (52,000× *g*) in a Beckman SW28 rotor for 15 h or 13.5 h at 4 °C. After centrifugation, equal-sized fractions (1.2 mL/fraction) were collected. The RNA from each fraction was isolated using TRIzol LS reagent (Invitrogen) and an equal volume of RNA from each fraction was used to synthesize cDNA using the SuperScript III First-Strand Synthesis SuperMix (Thermo Fisher Scientific, Waltham, MA, USA). The relative quantity of specific mRNAs was measured by reverse transcriptase (RT)- quantitative polymerase chain reaction (qPCR) (Thermofisher).

### 4.8. Nanoliposomal NH125 (NanoNH125)

The agent NH125 was encapsulated into a nanoliposome by first combining L-α-Phosphatidylcholine (ePC) and 1,2-Dipalmitoyl-sn-Glycero-3-Phosphoethanolamine-N-[Methoxy(Polyethylene glycol)-2000] ammonium salt (DPPE-PEG-2000) in chloroform at 80:20 mol % for a final lipid concentration of 25 mg/mL (Avanti Polar Lipids, Alabaster, AL, USA) [47,60]. 5.0 mg of NH125 (in methanol) was added to 1 mL of nanoliposome solution. The mixture was dried under nitrogen gas and re-suspended in saline at 60 °C. Following rehydration, the mixture was sonicated at 60 °C for 30 min followed by extrusion at 60 °C through a 100-nm polycarbonate membrane using an Avanti Mini Extruder (Avanti Polar Lipids). The particle size and charge were measured using a Malvern Zetasizer (Malvern Instruments, Malvern, UK), through procedures reported previously [47,60].

### 4.9. Characterization of NanoNH125

(a)Drug encapsulation. Efficiency of encapsulation of NH125 in the nanoliposomal formulation was estimated by UV-visible spectrophotometry (SPECTRAmax M2 plate reader; Molecular Devices, San Jose, CA, USA) using reported procedures [47,60,61]. Briefly, NanoNH125 solution was added to a 10 kDa Centricon filter tube (Millipore, Burlington, MA, USA) and centrifuged at 3750 rpm for 30 min to remove free NH125. Next, 0.5 mL of purified NanoNH125 was combined with 0.5 mL of a 1:1 solution of chloroform to methanol to destroy the nanoliposomal structure and release the drug into the solution. The lipids were separated via centrifugation at 10,000 rpm for 15 min. The supernatant was then used to measure NH125 concentration, calculated from a standard curve of NH125 from 0.01 to 1 mg/mL. The percentage of NH125 incorporated into nanoliposomes was calculated as incorporated NH125/total NH125 X 100 [47,60,61].(b)Stability. Stability of NanoNH125 stored at 4 °C was assessed weekly by comparing size and zeta potential change using the Malvern Zetasizer, using approaches reported previously [47,60,61].(c)Cell killing efficacy. Cell killing efficacy was measured by estimating the IC_50_ efficacy for killing UACC 903 cells by MTS assay and comparing these values to that of freshly manufactured NanoNH125, which is a standard published approach for assessing nanoparticle killing efficiency [47,60,61].(d)Assessment of 7-day repeated dose toxicity of NH125 and NanoNH125. The 7-day repeated dose toxicity was performed as described previously [47]. Swiss-Webster mice were treated with NH125 intraperitoneally or NanoNH125 intravenously at indicated doses daily for 7 days. Changes in the animal body weights, and behaviours were monitored daily.

### 4.10. Animal Studies

Tumour inhibition efficacy studies were performed in nude mice using procedures described previously [6,47,58,62]. Briefly, 1 million UACC 903 or 1205 Lu cells were injected in both flanks of 4–6-week-old female nude Balb/c mice. When the tumours were well-vascularized on day 7, animals were randomized and treated daily with 0.5 mg/kg NH125 intraperitoneally or 1–2 mg/kg NanoNH125, intravenously. Day 7 was selected, as histological assessment has confirmed the presence of a vascularized cell mass at this time point, which was determined to be a tumour. The DMSO and empty nanoliposomes were used as vehicle controls. Tumour volumes, animal weight and behaviour were monitored continuously on alternate days. Animals were sacrificed after tumour volumes in the vehicle control groups exceeded 1500 mm^3^ and tumours were subsequently collected and processed as needed. For the siRNA knockdown experiments, cells nucleofected with siScramble, siBRAF and siEEF2K were allowed to recover for 24 h in culture before subcutaneously injection into nude mice. For the LDL rescue experiment, 2, 4, and 6 mg LDL (in cottonseed oil) were administrated orally for 3 days before the cancer cell implantation and continuously throughout the experiment. All animal experiments were repeated twice or more times to include a minimum of 8 mice in each group. Animal experimentation was approved by the Animal Care and Use Committee of Pennsylvania State University and met National Institutes of Health standards as set forth in the “Guide for the Care and Use of Laboratory Animals” (Protocol # 46539-128690).

### 4.11. Statistics

Statistical analysis was undertaken using the one-way/two-way ANOVA GraphPad PRISM Version 7.04 software. Dunnett’s as post hoc analysis was performed when there was a significant difference. Results were considered significant at a *p*-value of <0.05.

## Figures and Tables

**Figure 1 ijms-23-03481-f001:**
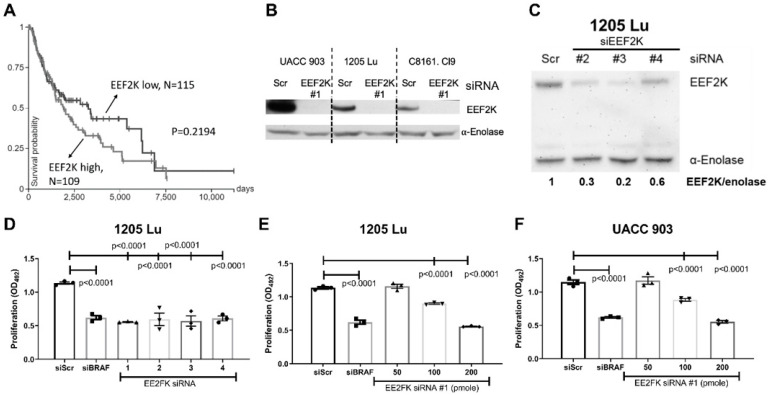
Knockdown of EEF2K inhibited melanoma cell proliferation. Data from the TCGA database suggest poorer survival with EEF2K overexpression (**A**) in melanoma patients. *n* = 115 for melanoma patients with low EEF2K; *n* = 105 for melanoma patients with high EEF2K. *p* = 0.2194. The data are available through the UCSC Xena Cancer Browser. Western blot showing knockdown of EEF2K with siRNA #1 in UACC 903, 1205 Lu and C8161Cl9 cell lines (**B**). Alpha-enolase served as the control for protein loading. Western blot showing knockdown of EEF2K with siRNAs #2, 3, 4 in UACC 903 (**C**). siRNA knockdown of EEF2K (siEEF2K) significantly reduced the growth of 1205 Lu (**D**) cells after 72 h in an MTS metabolism assay. Additionally, siRNA #1 dose dependently reduced the growth of 1205 Lu (**E**) and UACC 903 cells (**F**). siRNA to BRAF served as a positive control while scrambled (scr) siRNA served as the negative control. Significance was compared to scrambled knockdown by one-way ANOVA followed by Dunnett’s as post-hoc analysis. Experiments were replicated for *n* = 3 and representative graphs are shown.

**Figure 2 ijms-23-03481-f002:**
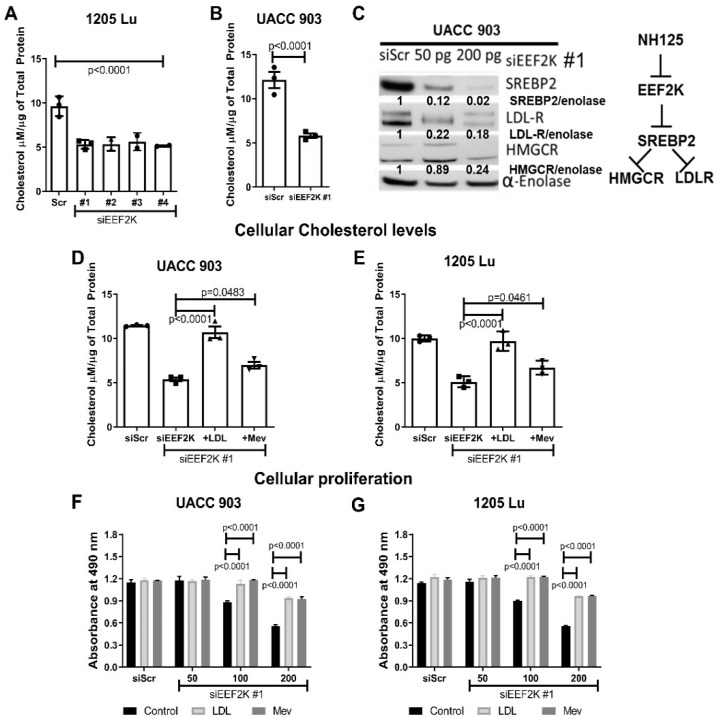
EEF2K modulated cholesterol metabolism in cancer cells. Genetic knockdown of EEF2K using siRNA (siEEF2K) reduced the cholesterol levels in 1205 Lu (**A**), UACC 903 (**B**) melanoma cell lines compared to scrambled siRNA knockdown. The cholesterol levels were normalized to the total protein content in cells. Significance was compared to scrambled siRNA knockdown (siScr) by *t*-test analysis (*n* = 3). Knockdown of EEF2K led to a dose dependent reduction of major cholesterol biomarkers LDLR, HMGCR and SREBP2 as measured by protein levels. Alpha-enolase served as a protein loading control (**C**). The cholesterol levels were rescued by the addition of LDL or mevalonic acid (Mev) to the media of both UACC 903 (**D**) and 1205 Lu cells (**E**). Significance was compared to scrambled knockdown (siScr) by one-way ANOVA followed by Dunnett’s as post-hoc analysis. Similarly, addition of LDL and mevalonic acid to the cell growth media rescued the proliferation of both UACC 903 (**F**) and 1205 Lu (**G**) cells. Significance was compared to scrambled knockdown (siScr) by two-way ANOVA followed by Dunnett’s as post-hoc analysis (*n* = 3).

**Figure 3 ijms-23-03481-f003:**
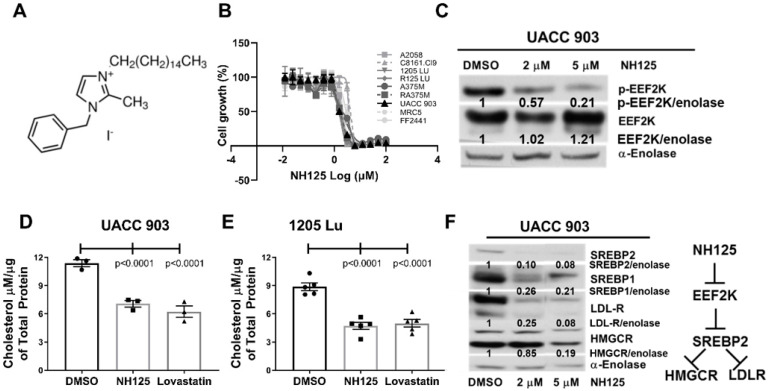
Pharmacological knockdown of EEF2K reduced cholesterol levels in melanoma. The pharmacological inhibitor of EEF2K, NH125 (**A**) significantly reduced the proliferation of melanoma cells after 72 h using an MTS assay (**B**) and decreased EEF2K phosphorylation (**C**) compared to total EEF2K levels. Alpha-enolase served as a protein loading control. Experiments were replicated three times. 5 µM NH125 reduced the cholesterol levels in UACC 903 (**D**) and 1205 Lu (**E**) cells compared to control DMSO. Significance was measured by one-way ANOVA followed by Dunnett’s as the post-hoc analysis. Lovastatin treatment served as a positive control for the inhibition of cholesterol synthesis (*n* = 3). Mechanistically, NH125 dose dependently lowered the LDLR, HMGCR, SREBP2 and SREBP1 levels in UACC 903 cells to reduce cholesterol levels (**F**).

**Figure 4 ijms-23-03481-f004:**
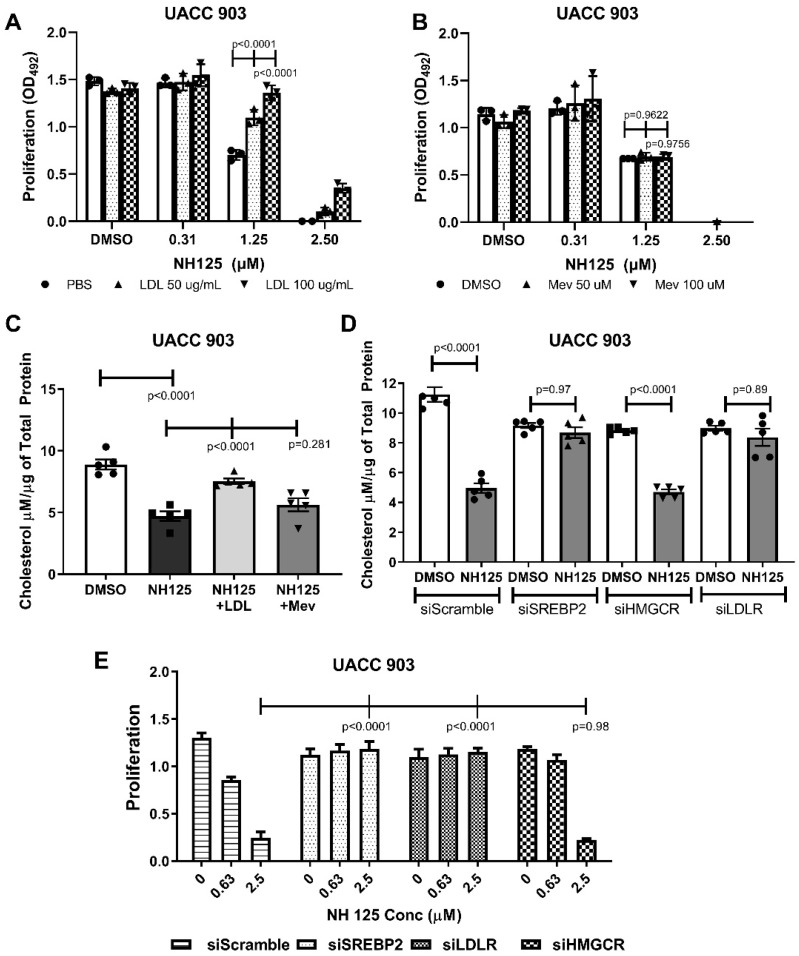
NH125 reduced cholesterol levels through regulation of SREBP2 and LDLR. The effect of NH125 on proliferation of melanoma cells could be completely rescued by addition of LDL (**A**) but addition of mevalonic acid (Mev) had a negligible effect (**B**). Significance was compared PBS/DMSO controls by two-way ANOVA followed by Dunnett’s as post-hoc analysis (*n* = 3). Similarly, the reduction of cholesterol levels following treatment with NH125 could be rescued by addition of LDL but not with the addition of mevalonic acid (**C**). Knockdown of SREBP2 and LDLR led to a decreased effect of NH125 on cholesterol levels (**D**) or proliferation (**E**) in UACC 903 melanoma cells. Significance was compared to the respective controls by one-way ANOVA followed by Tuckey’s as post-hoc analysis (*n* = 3).

**Figure 5 ijms-23-03481-f005:**
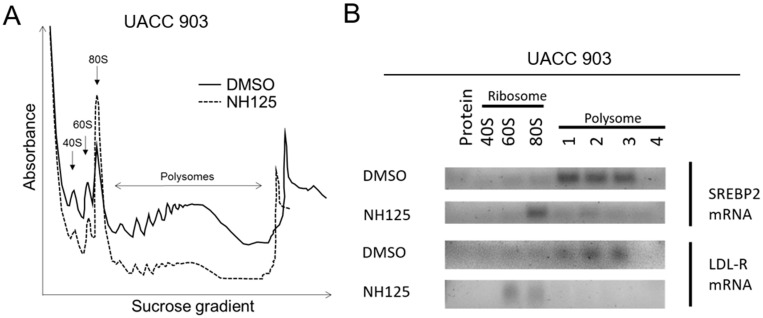
NH125 decreased the translation of SREBP2 and LDLR. Cell lysates were subjected to sucrose density gradient centrifugation to collect the ribosomal and polysomal fractions; RNA was extracted from pooled fractions; and analysed by PCR for SREBP2 as well as LDL-R mRNA abundance. The primers were designed using the Primer-BLAST software on the NCBI website. 5 µM NH125 reduced the polysomal fraction of UACC 903 cells after 24 hours of treatment with a concomitant increase in the 80S ribosomal fraction compared to DMSO treated cells (**A**). mRNAs for SREBP2 and LDLR shifted from the polysomal fraction to the ribosomal fraction after NH125 treatment (**B**). The experiments were replicated thrice and representative images are shown. The gel images were inverted to increase the resolution.

**Figure 6 ijms-23-03481-f006:**
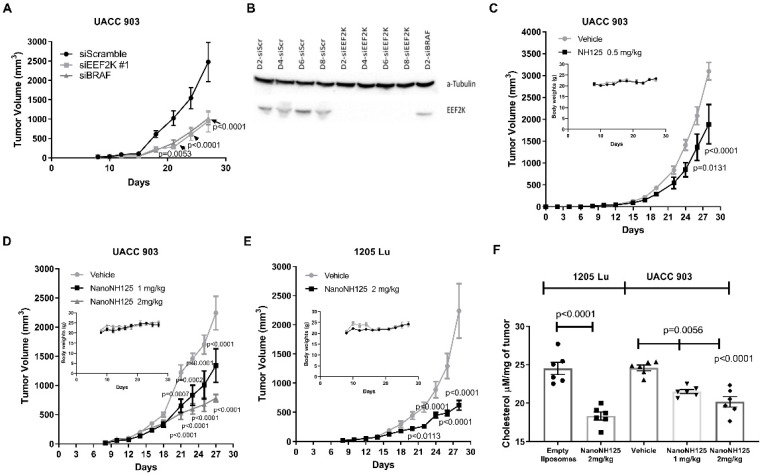
Targeting EEF2K by siRNA and NanoNH125 inhibited melanoma tumour growth through reduction of cholesterol levels. The siRNA-mediated knockdown of EEF2K reduced melanoma tumour development (**A**). The EEF2K siRNA knocked down UACC 903 cells were injected subcutaneously into nude mice and tumour growth kinetics were measured and compared to scrambled siRNA controls (*n* = 8). The significance at each time point was compared to scrambled siRNA (siScr) by two-way ANOVA followed by Dunnett’s as post-hoc analysis. BRAF knockdown served as a positive control (A). Western blots confirming the required knockdown of EEF2K for 8 days while alpha-tubulin served as loading control (**B**). NH125 inhibited tumour growth of UACC 903 xenografts compared to DMSO control by 34% when dosed at 0.5 mg/kg (*n* = 8) (**C**). The significance at each time point was compared to DMSO controls by two-way ANOVA followed by Dunnett’s as the post-hoc analysis. Higher doses of NH125 could not be tested because of toxicity. NanoNH125 significantly inhibited tumour growth of UACC 903 (**D**) and 1205 Lu (**E**) xenografts by 65% and 72%, respectively, compared to empty liposome vehicle control at 2 mg/kg following 28 days of treatment (*n* = 8). The significance at each time point was compared to DMSO controls by two-way ANOVA followed by Dunnett’s as the post-hoc analysis. NanoNH125 did not significantly affect animal body weight (**D**, **E**-insets) compared to empty liposome vehicle control. The NH125 treatment reduced cholesterol levels in tumours normalized by weight to tumours treated with empty liposome controls (**F**). To determine whether dietary supplementation with LDL can reverse the tumour inhibition mediated by siRNA-mediated EEF2K downregulation, 2 to 6 mg of LDL were orally administered to mice daily for three days prior to tumour cell implantation and continued daily for the duration of the experiment. Daily dietary supplementation of LDL slightly reversed the siEEF2K-mediated tumour inhibition (Appendix A). The increase in tumour growth with LDL supplementation was predicted to be subtle as these types of studies in animals are notoriously difficult to achieve. Since the reversal of the siEEF2K effect on tumour development was subtle, it may be due to insufficient pre-treatment or an inappropriate delivery route to achieve the optimal cholesterol levels needed for a full reversal.

**Figure 7 ijms-23-03481-f007:**
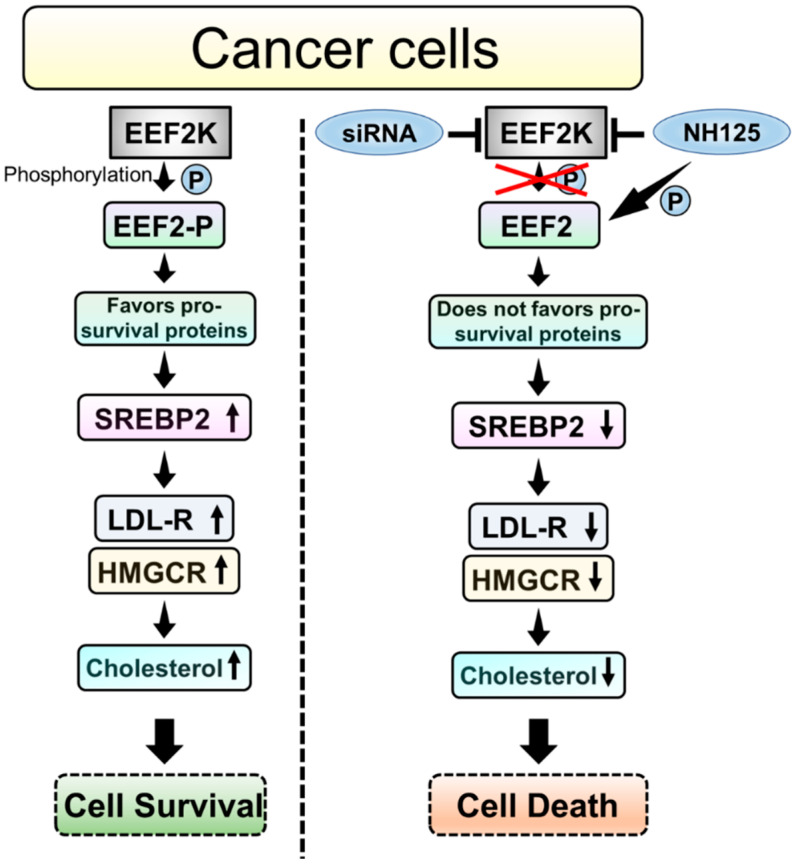
Schematics of mechanisms by which targeting EEF2K with siRNA or NH125 inhibit melanoma cell survival. Active EEF2K mediates protein translation through EEF2 and increases the levels of SREBP2, LDLR and HMGCR to increase cellular cholesterol and promote cell growth in cancer cells. Targeting EEF2K by siRNA and NH125 decreases the expressions of SREBP2, LDLR and HMGCR, which results in reduced cellular cholesterol and increased cell death.

## Data Availability

All data described in this article are available upon request. Please send the request to Gavin Robertson, gpr11@psu.edu.

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
