# Peer review of "Targeting Protein Translation in Melanoma by Inhibiting EEF-2 Kinase Regulates Cholesterol Metabolism though SREBP2 to Inhibit Tumour Development"

_ijms, 2022, doi:10.3390/ijms23073481_

Round 1

Reviewer 1 Report

  1. Your enolase in westerns is usually a bad presenter of loading contol, in some cases there's obviuos difference in loading, in other cases the control is too faint to be presented.
  2. figure 1, A ... since Kaplan Mayer does not look very significant ... therefore more explanation is needed, pleas provide p values, the numbers of patients in each curve, etc. 
  3. Mention p values in text as well, not only in figures
  4. There are several typos in text/figures, etc. 

Reviewer 2 Report

Targeting cholesterol metabolism as a putative treatment of cancer is not original, as many groups worldwide have been doing the same for various cancers. However, Dinavahi and colleagues have deciphered the effect of an inhibitor of EEF2K in the inhibition of melanoma cell growth. Besides, one of the most interesting point is that the authors have succeeded to decrease the toxicity of this drugs and hence to increase its efficacy in vivo with less deleterious side effects in an animal model.

The only weak point is that the authors have focused on the cholesterol uptake and its de novo synthesis. They have not explored cholesterol efflux (through the analysis of ABC protein accumulation) and storage. Numerous works pointed out the link between the level of ABCA1 and melanoma (cf. Oncotarget 2017). Besides, it is also usually classical to investigate the LXR-regulated pathway when SREBP2 (their mirror effect protein) is involved. Even if it is difficult to ask for all this experiments, at least these point should be discussed.

Minor points:

  • In the legend and in the main text, there is a recurrent mistake: "Figureure" instead of Figure.
  • Could the authors justify the use of a-enolase as house keeping protein instead of the "classical" b-actin? This point is important as the overexpression of a-enolase has been described in many cancers with poor diagnosis.
  • Higher doses of NH125 have been tested. However, have the authors tested lower doses to calculate an IC50?
  • Is NH125 efficient in melanoma cells only? Have the authors tried non melanoma derived cell lines such as PC3, HepG2, MDA...?
  • Does NH125 have an effect on the accumulation of SREBP1c and on the level of FASN, ACC, SCD and finally on fatty acid and/or TG homeostasis? This is also important for the survival of tumor cells.
  • What are the effects of a NanoNH125 treatment on both hepatic and circulating level of cholesterol and TG?

Round 2

Reviewer 2 Report

The authors have partially answered the reviewers's comments.

For points 4 and 5, the justification stating that some questions are "... outside of the scope of this manuscript" is somehow odd as performing one point analysis (e.g. on cell survival) on PC3 cells (or another cell type) is not that long and could have answered the question, the same for a qPCR for Fasn...

Regarding point 6, measuring " animal mortality, behaviors and weight" does not answer the important point of circulating cholesterol levels. 

Despite these comments, the manuscript quality has increased.
